# Interaction of preimplantation factor with the global bovine endometrial transcriptome

Ruth E. Wonfor[ID]*, Christopher J. Creevey[¤a], Manuela Natoli[¤b], Matthew Hegarty, Deborah M. Nash, Michael T. Rose[¤c]

Institute of Biological, Environmental and Rural Sciences, Aberystwyth University, Ceredigion, United Kingdom

¤a Current address: Institute for Global Food Security, Queen's University Belfast, Belfast, United Kingdom
¤b Current address: Cancer Research UK, Cambridge, United Kingdom
¤c Current address: Tasmanian Institute of Agriculture, University of Tasmania, Hobart, Australia
* rec21@aber.ac.uk

**Data Availability Statement:** The data has been uploaded to GEO under the series record GSE153699. Individual samples accession numbers are as follows: GSM4649298 Bovine

## Abstract

Preimplantation factor (PIF) is an embryo derived peptide which exerts an immune modulatory effect on human endometrium, promoting immune tolerance to the embryo whilst maintaining the immune response to invading pathogens. While bovine embryos secrete PIF, the effect on the bovine endometrium is unknown. Maternal recognition of pregnancy is driven by an embryo-maternal cross talk, however the process differs between humans and cattle. As many embryos are lost during the early part of pregnancy in cattle, a greater knowledge of factors affecting the embryo-maternal crosstalk, such as PIF, is needed to improve fertility. Therefore, for the first time, we demonstrate the effect of synthetic PIF (sPIF) on the bovine transcriptome in an *ex vivo* bovine endometrial tissue culture model. Explants were cultured for 30h with sPIF (100nM) or in control media. Total RNA was analysed via RNA-sequencing. As a result of sPIF treatment, 102 genes were differentially expressed compared to the control ($P$adj<0.1), although none by more than 2-fold. The majority of genes (78) were downregulated. Pathway analysis revealed targeting of several immune based pathways. Genes for the TNF, NF-κB, IL-17, MAPK and TLR signalling pathways were down-regulated by sPIF. However, some immune genes were demonstrated to be upregulated following sPIF treatment, including *C3*. Steroid biosynthesis was the only over-represented pathway with all genes upregulated. We demonstrate that sPIF can modulate the bovine endometrial transcriptome in an immune modulatory manner, like that in the human endometrium, however, the regulation of genes was much weaker than in previous human work.

## Introduction

The embryo preimplantation period is complex; it involves modulation of the maternal uterine immune response and acceptance of the embryo, and embryo-maternal cross talk is essential to the process. Preimplantation factor (PIF) is a peptide secreted by viable embryos as early as the two-cell stage, identified in human, murine, bovine and porcine models [1, 2]. Secretion of

endometrium_1_Control, GSM4649299 Bovine
endometrium_1_sPIF, GSM4649300 Bovine
endometrium_2_Control, GSM4649301 Bovine
endometrium_2_sPIF, GSM4649302 Bovine
endometrium_3_Control, GSM4649303 Bovine
endometrium_3_sPIF, GSM4649304 Bovine
endometrium_4_Control, GSM4649305 Bovine
endometrium_4_sPIF, GSM4649306 Bovine
endometrium_5_Control, GSM4649307 Bovine
endometrium_5_sPIF, GSM4649308 Bovine
endometrium_6_Control, GSM4649309 Bovine
endometrium_6_sPIF, GSM4649310 Bovine
endometrium_7_Control, GSM4649311 Bovine
endometrium_7_sPIF.

**Funding:** R. Wonfor was funded by Aberystwyth University through the Doctoral Career Development Scheme. The funders had no role in study design, data collection and analysis, decision to publish, or preparation of the manuscript.

**Competing interests:** No authors have competing interests.

PIF from murine embryos in culture is greater at the blastocyst development stage, compared to the morula, demonstrating a role of PIF both in early and later developmental stages of the preimplantation conceptus [1]. Furthermore, PIF has been detected in bovine serum at 20 days post fertilisation [3]. Human embryos that do not secrete PIF fail to implant, thus underpinning the importance of PIF in the embryo-maternal dialogue at the implantation stage [4]. In humans, PIF modulates the maternal uterine immune response which aids the acceptance of the embryo [2]. Synthetic PIF (sPIF) interacts with decidualized human endometrial stromal cells and first trimester decidual cells through three specific pathways: immune tolerance, embryo adhesion and apoptosis/remodelling of the uterus, all of which are fundamental to embryo implantation and maternal recognition of pregnancy [5]. Furthermore, sPIF targets naïve CD14+ peripheral blood mononuclear cells and reduces secretion and mRNA expression of Th1/Th2 cytokines [6, 7]. In addition, sPIF modulates the uterine immune response to aid in embryo acceptance by promoting a Th2 bias and inducing an anti-inflammatory effect, whilst also preserving Th1 responses necessary for protecting the mother from invading pathogens [5, 6, 8, 9].

Interferon-τ (IFN-τ) is a well characterised, crucial embryo derived signal. Bovine conceptus secretion of IFN-τ begins around formation of the trophectoderm and peaks between day 15 and 17 of pregnancy, when the conceptus is an elongated filamentous structure, which instigates maternal recognition of pregnancy in ruminants and thus, early pregnancy establishment [10–13]. However, secretion of IFN-τ rapidly declines from day 21 onwards [13]. It is clear that IFN-τ is imperative for the embryo-maternal crosstalk and modulation of the endometrial immune profile [14], however, the establishment and recognition of pregnancy is more complex than the presence of IFN-τ alone [10, 15, 16]. As the fertility of dairy cows has declined in recent years, and a considerable proportion of pregnancy losses occur during early pregnancy [11, 17], it is imperative to understand this critical window to improve fertility rates in cattle.

Several attempts have aimed at understanding the bovine preimplantation embryo-maternal crosstalk on a global transcriptome level [10, 16, 18–23]. The dynamic modulation of the maternal immune system is essential to aid in implantation, growth of the embryo and ultimately, a successful pregnancy [24, 25]. The bovine preimplantation embryo has clear roles in modulating endometrial gene expression, to both suppress the immune response for promotion of maternal embryo tolerance, whilst also increasing innate immune related genes to prevent vulnerability of the uterine environment to pathogens [19, 26]. Thus, there is the potential that PIF may be involved in this cross talk.

Although it is known that PIF is secreted by viable bovine embryos and detectable in bovine serum through early pregnancy [1, 3], there is currently limited evidence of any effect of PIF on maternal bovine tissue and in the embryo-maternal crosstalk. We have previously reported that sPIF reduces native IL-6 secretion *in vitro* from non-pregnant bovine endometrial tissue during the early luteal and follicular stage of the oestrous cycle [27]. We report here for the first time the effect of sPIF on the native endometrial global transcriptome through RNA-sequencing. Synthetic PIF is hypothesised to have an immune modulatory role in cattle, similar to that described in the human. Although, due to differences in the maternal recognition of pregnancy and the timings and mode of implantation in humans compared to cattle, it was deemed likely that there would be some differences in the role of PIF between these species.

## Materials and methods

### Animals

Bovine uteri (n = 7) and corresponding blood samples were collected from heifers presented for slaughter at a local abattoir. As post-slaughter material was used, licencing through the

Animals (Scientific Procedures) Act 1986 and ethical review were not necessary. Based on previous work [27], uteri with stage IV ovaries were investigated to allow the study of sPIF on endometrial tissues that were not under the immune suppressive effects of progesterone [28, 29]. Samples were staged by assessing ovarian morphology as previously described [30, 31]. Briefly, stage IV was defined as having a regressing corpus luteum with a diameter of < 1 cm [30]. To ensure there was no underlying inflammation in the sampled tissue, cytology samples were taken from the endometrium at the abattoir, using a modified cytobrush technique, and assessed for percentage of polymorphonuclear cells (PMN), as previously described [27]. A threshold of PMN percentage greater than 5% was set to exclude animals based on the guideline of detection of subclinical endometritis [32, 33], although all samples were below 5% PMN and therefore none were excluded.

Uteri and blood samples were stored on ice during the one-hour transportation back to the laboratory. Tissues were used for explant culture and blood serum for serum progesterone concentration via ELISA (DRG Diagnostics, Marburg, Germany). To support ovarian morphology staging, the blood sera were used for progesterone analysis. Samples were deemed to have high progesterone if serum concentrations were above 1 ng/mL [34]. Based on this threshold, samples were split into a high and low progesterone group. The limit of detection of the progesterone assay was 0.01 ng/mL and the intra-assay CV was 5.5%.

## Endometrial explant tissue culture

Tissue culture was established using the method described by Borges *et al.* [35]. Briefly, endometrial tissue was sampled randomly from intercaruncular tissue in the first third (closest to the utero-tubular junction) of the uterine horn ipsilateral to the staged ovary, using an 8 mm biopsy punch. The endometrial tissue was then dissected away from the myometrium using sterile scissors. Six biopsies were taken per animal. Samples were weighed (mean ± SD weight was 42.47 ± 7.7 mg) and one biopsy placed per well in 6 well plates (Corning, Amsterdam, The Netherlands) with 3 mL of RPMI 1640 media (Gibco, Life Technologies, Paisley, UK) supplemented with 50 IU/mL penicillin, 50 µg/mL streptomycin (Sigma-Aldrich, St. Louis, MO, USA) and 2.5 µg/mL amphotericin B (Sigma-Aldrich). Explants were incubated in a sterile incubator at 37 $^{\circ}$C and 5% $CO_2$ for 30 h.

Synthetic PIF (MVRIKPGSANKPSDD) was synthesised with > 95% purity by Bioincept (New Jersey, USA). The amino acid structure of the human 15 amino acid PIF has previously been analysed and the 3D structure predicted [6]. The sPIF used in the present study was identical to that used in all other published research on the peptide.

Whole explant biopsies from each animal were treated with either medium alone or with sPIF (100nM) for 24 h in 6 well plates. As DMSO was used in the reconstitution of sPIF, the same amount of DMSO was added to the control wells. Based on our previously described methodology [27], following the 24 h incubation medium was removed and replaced with fresh medium containing the same treatments for another 6 h. At the end of the 30 h period, explants were stored individually in 1 mL RNAlater (Invitrogen, Life Technologies, Paisley, UK) at 4 $^{\circ}$C for 24 h. The RNAlater was then removed and explants stored at -80 $^{\circ}$C until further processing.

## Total RNA extraction

Total RNA was extracted from two explants (one for each treatment, control or sPIF) per animal, using the Total RNA purification plus kit (Norgen Biotek Corp., Ontario, Canada), to give a total of 14 samples of RNA. Briefly, from each explant that RNA was to be extracted from, ≤ 20 mg of tissue was cut off whilst still frozen, using sterile scissors and placed in the manufacturer's lysis buffer. Samples were then subsequently subjected to bead beating, to aid

tissue disruption, whereby a 5 mm stainless steel bead (Qiagen, Manchester, UK) was added and samples placed in a TissueLyser (Qiagen, Manchester, UK) for 2 minutes at 50 oscillations per second. Samples were centrifuged at 14,000 x g for 1 minute to pellet any remaining debris and the supernatant extracted according to the manufacturer's instructions.

The quality of all RNA extracted was assessed with a 2100 Bioanalyzer (Agilent Technologies, Santa Clara, USA) following the method described by the manufacturer. Samples were of suitable quality, showing RNA integrity numbers (RIN) above 7.

## RNA-sequencing library preparation and next generation sequencing

Following assessment of RNA quality, all samples were prepared for sequencing and sequenced at the Translation Genomics facility in IBERS, Aberystwyth University. Total RNA samples were prepared for sequencing using the TruSeq v2 kit (Illumina, San Diego, USA), using the manufacturer's protocol, up to the validate library step. Following the enrichment of cDNA fragments with adapters, the cDNA was quantified using a Qubit 2.0, dsDNA broad range assay (Invitrogen), following the protocol supplied by the manufacturer. Each sample was diluted to 10 nM with 10 nM Tris HCl and 0.5% Tween-20 in nuclease free water. The use of adapters allowed multiple indexing of samples and so, samples were pooled and subsequently diluted to 2 nM with elution buffer (Qiagen, Manchester, UK) and then to 1 nM with 0.1 M sodium hydroxide, before being held at room temperature for 5 minutes to denature the DNA. Following denaturation, the samples were diluted to 10 pM in hybridisation buffer and loaded onto a cBot (Illumina, San Diego, USA) to cluster cDNA onto the Flow cell. The 14 samples were clustered onto 2 lanes of a V4 High output flow cell and subsequently paired-end sequenced on a HiSeq 2500 (Illumina, San Diego, USA). Base pairs (bp) per read were set to 126 bp. Six samples (from 3 cows) on one lane were sequenced twice due to a sample loading error, which resulted in low reads compared to the 8 samples on the other lane. Both reads were included in the subsequent sequencing analysis pipeline and were processed separately until after the featureCounts step.

## Sequencing analysis pipeline

A previously described RNA-seq pipeline was adapted for use in the study [36]. All work up to the statistical analysis was completed on the open source platform Galaxy [37–39], hosted by IBERS, Aberystwyth University.

**Read quality assessment and trimming.** Raw paired-end data were submitted to FastQC analysis (Galaxy version 0.69; Babraham Bioinformatics). Based on the quality of the reads outlined by FastQC, samples were trimmed using Trimmomatic [40], utilising an initial Illumina-clip, headcrop, crop and Minlen (to remove any reads below 50 bp) functions. The quality of the resulting paired-end data was again assessed via FastQC.

**Alignment to the bovine genome.** Bowtie 2 (Galaxy version 2.2.6) [41, 42] was used to map reads to the reference bovine genome. Samples were mapped to the UMD 3.1 assembly of the *Bos taurus* genome from Ensembl (version 89; http://www.ensembl.org/).

**Gene expression data and statistical analysis.** Read abundance for annotated genes was calculated using the featureCounts package (Galaxy version 1.4.6.p5) [43]. Reads for the two sequencing runs for six samples were joined together after the featureCounts step by adding together the raw counts for each gene from each run. The raw counts of sequencing reads generated by featureCounts were used for all statistical analyses.

The Bioconductor package deSeq2 was used to determine the differential expression of genes as part of the R software package (version 3.4.0) [44]. Prior to the statistical modelling, deSeq2 analysis removed any genes that had less than 10 counts for any one sample. The statistical model was set to recognise that all samples were paired, with control and sPIF treated

explants originating from the same animal. This was completed by running a multifactorial design, thus controlling for extra variation in the data set and subsequently improving the sensitivity of the analysis. To assess if the effect of sPIF treatment differed between lanes, lane was added into the data frame as a factor and interaction terms used. The same interaction terms analysis was completed for serum progesterone concentration, with samples being split into a high and low progesterone as described, to determine if the effect of sPIF differed between progesterone groups. To determine significant differentially expressed genes (DEG), the P adjusted value $P$adj$<0.1$ was used, based on the Benjamini-Hochberg false discovery rate [45].

To determine if DEGs were involved in separate biological functions and pathways, gene ontology (GO) categories and KEGG pathways [46] were investigated using STRING (version 11.0) and the genes used in the DESeq analysis used as the statistical background [47]. P adjusted values for over-represented GO categories and over-represented KEGG pathways were identified and significance set at $P$adj$<0.05$.

Protein to protein interactions within the DEGs network were evaluated using STRING (version 11.0) and the *B. taurus* genome used as the statistical background [47]. Initially all prediction methods within the STRING analysis were used (neighbourhood, gene fusion, co-occurrence, co-expression, experiment databases and textmining), however due to the discovery of several non-specific results, the textmining prediction was subsequently removed, which demonstrated a more focussed network.

## Results

### Progesterone concentration and endometrial cytology

Progesterone concentrations were below 1ng/mL in three animals and were therefore assigned to a low progesterone group, with a mean concentration of 0.69 ng/mL ± 0.06 (standard error of the mean) and a range of 0.59–0.81 ng/mL. The remaining four animals had progesterone concentrations greater than 1 ng/mL and were assigned to a high progesterone group, with a mean concentration of 3.1 ng/mL ± 0.86 (standard error of the mean) and a range of 1.44–5.41 ng/mL. There was no evidence of subclinical inflammation in any of the uterine samples, with < 5% PMN in all cytobrush smears.

RNA-sequencing overview

RNA-sequencing resulted in a total of 245,924,502 million paired-end reads across all fourteen samples. Following mapping of the reads to the reference genome *B. taurus* UMD3.1, 15,681 transcripts were analysed for differential expression in the bovine endometrial tissue samples following sPIF treatment. The overall mean counts for each gene included in the differential expression analysis was 1,272.9 counts ± 5,885.6 (standard deviation). The mean gene count data were skewed with the majority of genes (74.7%) having under 10,000 counts, however the majority (58.8%) of DEGs were also located within this range of count data. To ensure that there was no difference in the two sequencing runs, mean counts and variability was assessed. There was limited difference between mean counts for genes included in the differential gene analysis between the samples from the two different RNA-seq lanes, with 1,231.9 counts ± 5,212.5 (standard deviation) and 1,302.6 counts ± 7,009.7 (standard deviation) for cows 1–4 (lane 1) and cows 5–7 (2 sequencing runs summed together), respectively. Furthermore, PCA analysis on the data prior to the two sequencing runs for cows 5–7 being summed together, demonstrated that the technical replicates for each sample clustered together and so were appropriate for combination in the analysis (S1 Fig).

### Sample variability

Variability between animal replicates and individual samples was assessed. There was a strong effect of animal replicates on the dataset variability, more so than sPIF treatment (Fig

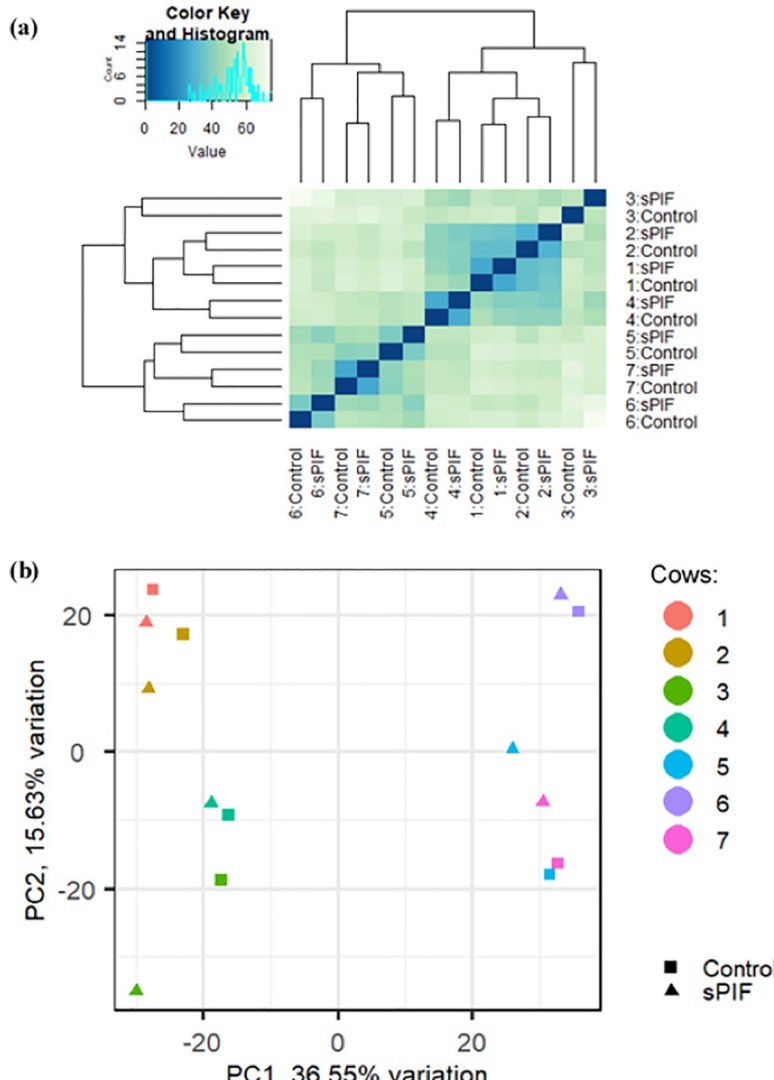

**Fig 1. Large variances were detected between samples.** (a) Heat map depicting the Euclidian distances between individual animal replicates and samples treated with or without sPIF (100nM), calculated from the regularised log transformation. (b) PCA plot showing the variance between individual animal replicates and samples treated with or without sPIF (100nM) in the first two principle components.

1). The heat map (Fig 1A) and PCA (Fig 1B) show clear differentiation between the samples in each lane (cows 1 to 4 lane 1, cows 5 to 7 lane 2), although there was no significant effect of lane on DEGs in the data set. When principle component (PC) 1 was compared against PC2, 3 and 4 it was noted that there was a clear grouping of samples from the different lanes, but this was not evident when PC1 was not included in the PCA plots (Fig 1B and S2 Fig). It is clear from Fig 1B that the variability was not attributed to serum progesterone concentration (High progesterone group cows 2–5; Low progesterone group cows 1, 6–7). As PC1 and PC2 only accounted for 52.2% of the variation, PC 1–4 were further examined (PC1, PC2, PC3 and PC4 accounted for 36.6%, 15.6%, 13% and 8.5% of the variation, respectively). However, none demonstrated a clear clustering of the low or high progesterone groups (S2 Fig).

## Identification of differentially expressed genes

Synthetic PIF treatment induced differential expression of 102 genes in bovine endometrial tissue explants ($P$adj<0.1; of which 33 were differentially expressed $P$adj<0.05); 78 of which were down-regulated and 24 up-regulated. No genes were up- or down-regulated greater than two-fold change. The full list of differentially expressed genes (DEG) is displayed in the supplementary material (S1 Table). The top 10 most significantly DEGs are displayed in Table 1. Two genes involved in immune pathways were among the most significantly downregulated genes following sPIF treatment (Table 1; *NFKB1*; $P$adj = 4.7 x $10^{-3}$ and *IRF1*; $P$adj = 5.8 x $10^{-3}$). There was no effect of lane or serum progesterone concentration on the whole data set nor an interaction with sPIF treatment ($P$adj>0.1), thus all differences found within this study were attributed to sPIF treatment.

## Gene ontology analysis

The biological pathway plasma membrane was over-represented with DEGs, following sPIF treatment which is within the cellular component ontology ($P$adj<0.05). The 'plasma membrane' (GO:0005886) was over-represented with the 17 DEGs ($P$adj = 0.013; *ADA*, *CALCRL*, CD40, *EMP3*, *GJC1*, *GNA14*, ICAM1, *IFNAR2*, *LPL*, *PTGDR*, *RAB8B*, *RGS16*, *RGS2*, *RHOF*, *SLC1A5*, *SLC34A2*, *TSPAN5*).

## Pathway analysis

A total of forty KEGG pathways were over-represented with DEGs, following sPIF treatment ($P$adj<0.05). Pathways were organised into biological categories using the KEGG BRITE Functional Hierarchies database, organising each pathway into a class and subclass (S2 Table). The overrepresented pathways fitted into six classes, Human diseases; Environmental information processing; Organismal systems; Metabolism; Cellular processes and Genetic information processing (S2 Table). Twenty-two pathways were classed as 'Human disease' pathways, largely due to DEGs involved in the NF-κB and TNF signalling pathways and the immune gene *C3*. As such, these pathways were discarded as they were deemed irrelevant to the dataset. A further pathway was discarded, 'Osteoclast differentiation', which was in the class 'Organismal Systems' and subclass 'Development', as it was irrelevant for the tissue studied and appeared as

**Table 1. Top 10 most significantly DEGs from the control, following sPIF treatment.**

| Gene ID | Gene symbol | Gene name | Log2 fold change | FDR* |
|---|---|---|---|---|
| *Under-expressed by sPIF* | | | | |
| ENSBTAG00000013705 | *NFKBIE* | NFKB inhibitor epsilon | -0.635 | 1 x $10^{-4}$ |
| ENSBTAG00000012178 | *NR1D1* | nuclear receptor subfamily 1 group D member 1 | -0.87 | 3.4 x $10^{-4}$ |
| ENSBTAG00000012343 | *TSPAN5* | tetraspanin 5 | -0.768 | 3.7 x $10^{-3}$ |
| ENSBTAG00000020270 | *NFKB1* | nuclear factor kappa B subunit 1 | -0.573 | 4.7 x $10^{-3}$ |
| ENSBTAG00000031231 | *IRF1* | interferon regulatory factor 1 | -0.767 | 5.8 x $10^{-3}$ |
| ENSBTAG00000011207 | *CNN1* | calponin 1 | -0.684 | 8.3 x $10^{-3}$ |
| *Over-expressed by sPIF* | | | | |
| ENSBTAG00000014149 | *LCN2* | lipocalin 2 | 1.177 | 1.4 x $10^{-4}$ |
| ENSBTAG00000018843 | *SERPINA1* | serpin family A member 1 | 1.203 | 1.8 x $10^{-3}$ |
| ENSBTAG00000009725 | *AOX1* | aldehyde oxidase 1 | 0.51 | 7.4 x $10^{-3}$ |
| ENSBTAG00000016255 | *PLEK2* | pleckstrin 2 | 0.655 | 7.4 x $10^{-3}$ |

* Based on P adjusted values (False discovery rate: FDR; $P$adj<0.1) as assessed by the Bioconductor package, deSeq2 statistical analysis.

over-represented again due to DEGs involved in the TNF and NF-κB signalling pathways. Once these irrelevant pathways were removed, a total of seventeen KEGG pathways were deemed relevant to the dataset (Table 2), which fitted into five KEGG BRITE Functional Hierarchies classes. Within these four classes, the Organismal Systems class and the subclass 'Immune system' had the greatest number of over-represented KEGG pathways in the dataset (seven pathways). The TNF (Fig 2) and NF-κB (Fig 3) signalling pathways, both of the Environmental Information Processing class and Signal transduction subclass, were highly significantly over-represented following sPIF treatment ($P$adj = 9.8 x $10^{-7}$; 5.5 x $10^{-7}$, respectively), with all genes in each pathway downregulated. The importance of these pathways within the whole dataset was clear due to the central signalling roles in a number of over-represented biological pathways, such as the IL-17, MAPK and TLR signalling pathways (Table 2), and explained the over-representation of a number of disease and infection pathways, which rely on these signalling pathways. Therefore, there was a clear indication of downregulation of immune factors following sPIF treatment, although it was noted that the complement component *C3* gene expression was upregulated (Log2 fold change 0.59; $P$adj = 0.09). Steroid biosynthesis was the only pathway with all DEGs upregulated (*CYP24A1*, *DHCR7*, *SQLE*; $P$adj = 3.5 x $10^{-3}$).

## Protein interaction networks

Known and predicted protein interactions within the DEGs dataset were analysed using STRING. All defined prediction methods were used apart from textmining (neighbourhood,

**Table 2. Relevant KEGG pathways significantly over-represented following sPIF treatment.**

| KEGG pathway | Number of DEGs | Observed DEGs | | FDR* |
|---|---|---|---|---|
| | | Downregulated | Upregulated | |
| TNF signalling pathway | 10 | *CSF1*, *CXCL3*, *VCAM1*, *ICAM1*, *MAPK11*, *NFKB1*, *TNFAIP3*, *TRAF1*, *TRAF2* | | 3.8 x $10^{-7}$ |
| NF-kappa B signalling pathway | 8 | *CD40*, *VCAM1*, *ICAM1*, *NFKB1*, *NFKB2*, *TNFAIP3*, *TRAF1*, *TRAF2* | | 5.5 x $10^{-7}$ |
| IL-17 signalling pathway | 6 | *CXCL3*, *MAPK11*, *NFKB1*, *TNFAIP3*, *TRAF2* | *LCN2* | 5.8 x $10^{-4}$ |
| Steroid biosynthesis | 3 | | *CYP24A1*, *DHCR7*, *SQLE* | 4 x $10^{-3}$ |
| NOD-like receptor signalling pathway | 6 | *CXCL3*, *IFNAR2*, *MAPK11*, *NFKB1*, *TNFAIP3*, *TRAF2* | | 4.6 x $10^{-3}$ |
| Prolactin signalling pathway | 4 | *IRF1*, *MAPK11*, *NFKB1*, *STAT5A* | | 8.2 x $10^{-3}$ |
| Cell adhesion molecules (CAMs) | 4 | *CD40*, *VCAM1*, *ICAM1* | *NEO1* | 9.1 x $10^{-3}$ |
| Necroptosis | 5 | *HIST1H2AC*, *IFNAR2*, *STAT5A*, *TNFAIP3*, *TRAF2* | | 0.01 |
| MAPK signalling pathway | 7 | *CSF1*, *GADD45G*, *IGF2*, *MAPK11*, *NFKB1*, *NFKB2*, *TRAF2* | | 0.01 |
| Th1 and Th2 cell differentiation | 4 | *MAPK11*, *NFKB1*, *NFKBIE*, *STAT5A* | | 0.01 |
| Toll-like receptor signalling pathway | 4 | *CD40*, *NFKB1*, *MAPK11*, *IFNAR2* | | 0.02 |
| Leukocyte transendothelial migration | 3 | *VCAM1*, *ICAM1*, *MAPK11* | | 0.02 |
| Protein processing in endoplasmic reticulum | 5 | *DNAJB1*, *ERO1B*, *HYOU1*, *PPP1R15A*, *TRAF2* | | 0.02 |
| Th17 cell differentiation | 4 | *MAPK11*, *NFKB1*, *NFKBIE*, *STAT5A* | | 0.02 |
| RIG-I-like receptor signalling pathway | 3 | *MAPK11*, *NFKB1*, *TRAF2* | | 0.03 |
| Adipocytokine signalling pathway | 3 | *NFKB1*, *NFKBIE*, *TRAF2* | | 0.04 |
| Apoptosis | 4 | *GADD45G*, *NFKB1*, *TRAF1*, *TRAF2* | | 0.04 |

*Based on P adjusted values (False discovery rate: FDR; $P$adj<0.05) as assessed by STRING analysis.

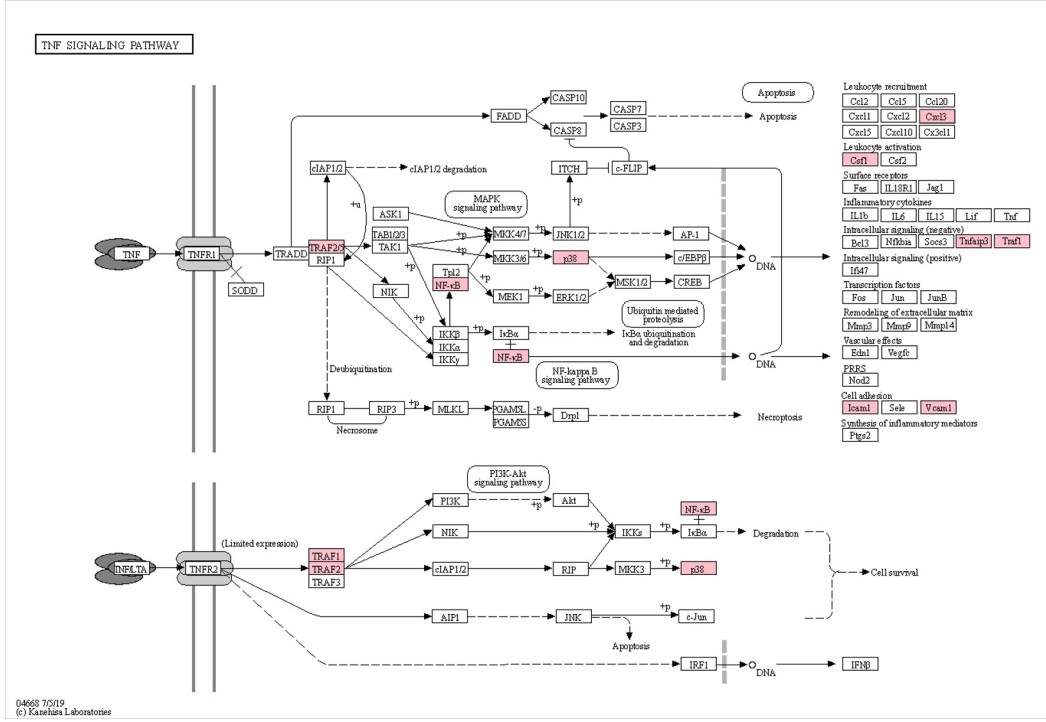

**Fig 2. Putative changes in the TNF signalling pathway induced by sPIF treatment.** Red boxes are proteins encoded for by DEGs, with reduced expression following sPIF treatment, as identified by STRING analysis, based on P adjusted values ($P$adj = 3.8 x 10$^{-7}$).

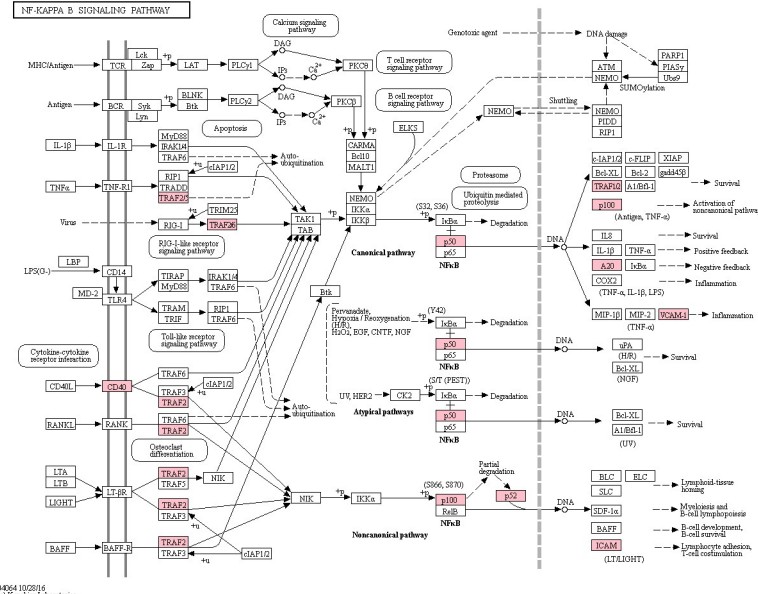

**Fig 3. Putative changes in the NK-κB signalling pathway induced by sPIF treatment.** Red boxes are proteins encoded for by DEGs, with reduced expression following sPIF treatment, as identified by STRING analysis, based on P adjusted values ($P$adj = 5.5 x 10$^{-7}$).

gene fusion, co-occurrence, co-expression and experiment databases). The overall network was significantly enriched ($P$adj = 9.24 x10$^{-9}$; *B. taurus* genome used as background gene list) with a total of 40 edges signifying connections between 34 proteins transcribed by DEGs following sPIF treatment. Fig 4 displays the proteins that are connected within the network of DEG following sPIF treatment and demonstrates which associations are stronger through the thickness of the edges between nodes. It was noted that there was a strong interaction network between NF-κB and TNF signalling related proteins (Fig 4).

## Discussion

This is the first study to demonstrate the effect of sPIF on the global endometrial bovine transcriptome. The investigation showed interaction of sPIF with the bovine endometrium, specifically that 102 genes were differentially expressed following sPIF treatment, with the majority (78 of 102 DEGs) downregulated. Furthermore, pathway analysis demonstrated sPIF to work in an immune modulatory manner on the bovine endometrium, as originally hypothesised. However, in the present study, no genes were modulated greater than two-fold following sPIF treatment. Thus, the bovine endometrial response to sPIF was much weaker than that demonstrated in decidualized human endometrial stromal cells and first trimester decidual cells, where some genes were modulated as much as 53 fold following sPIF treatment [5, 48, 49].

The present study used an *ex vivo* tissue explant method to model the effects of sPIF on the bovine endometrium. The use of whole tissue samples allowed assessment of sPIF in a model which maintains the tissue architectures of the endometrium, more akin to an *in vivo* state [35]. However, it is accepted that the *ex vivo* model likely adds variability into the dataset without a characterisation of populations of epithelial and stromal cells within each sample. Assessing the response on the whole tissue may partially explain the weaker response to sPIF in the bovine endometrium, compared to that demonstrated in individual cell types in humans [5, 48, 49]. Indeed, sPIF may have differing effects on bovine endometrial epithelial and stromal cells, and this warrants further study. However, a recent study used a similar methodology to assess the effect of bovine conceptuses and IFN- τ on the bovine endometrium, without characterising the populations of epithelial and stromal cells within each sample [16]. Therefore, in this study, we present the effect of sPIF on the whole bovine endometrial tissue structure.

We note that analysis of gene transcription alone does not account for possible post transcriptional changes that alter protein expression of the DEGs following sPIF treatment. Thus, further functional experiments, such as an assessment of the proteome, are needed to verify the effect of sPIF on the bovine endometrium in pregnancy. Furthermore, the present study set out to assess the general effect of sPIF on the bovine endometrial transcriptome, but assessing the effect of sPIF alone *in vitro* ignores the effect of other mediators within the uterine environment that may be maternal or conceptus derived. Therefore, the effect of other mediators in bovine pregnancy, such as IFN-τ, must be considered to fully understand the relationship with PIF and bovine pregnancy.

### Variation between animal replicates

Variation between animal replicates had a strong effect on the data set and more so than that of sPIF treatment. It is acknowledged that a difficulty in endometrial transcriptome studies is the variability introduced by animal status and management [50]. Indeed, increased progesterone levels can alter the endometrial transcriptome in heifers during early pregnancy [51]. However, despite some samples having higher than expected serum progesterone concentrations, indicating that they were in the luteal phase, there was no effect on the data set in the present study. Lactation status has been shown not to affect endometrial gene expression in

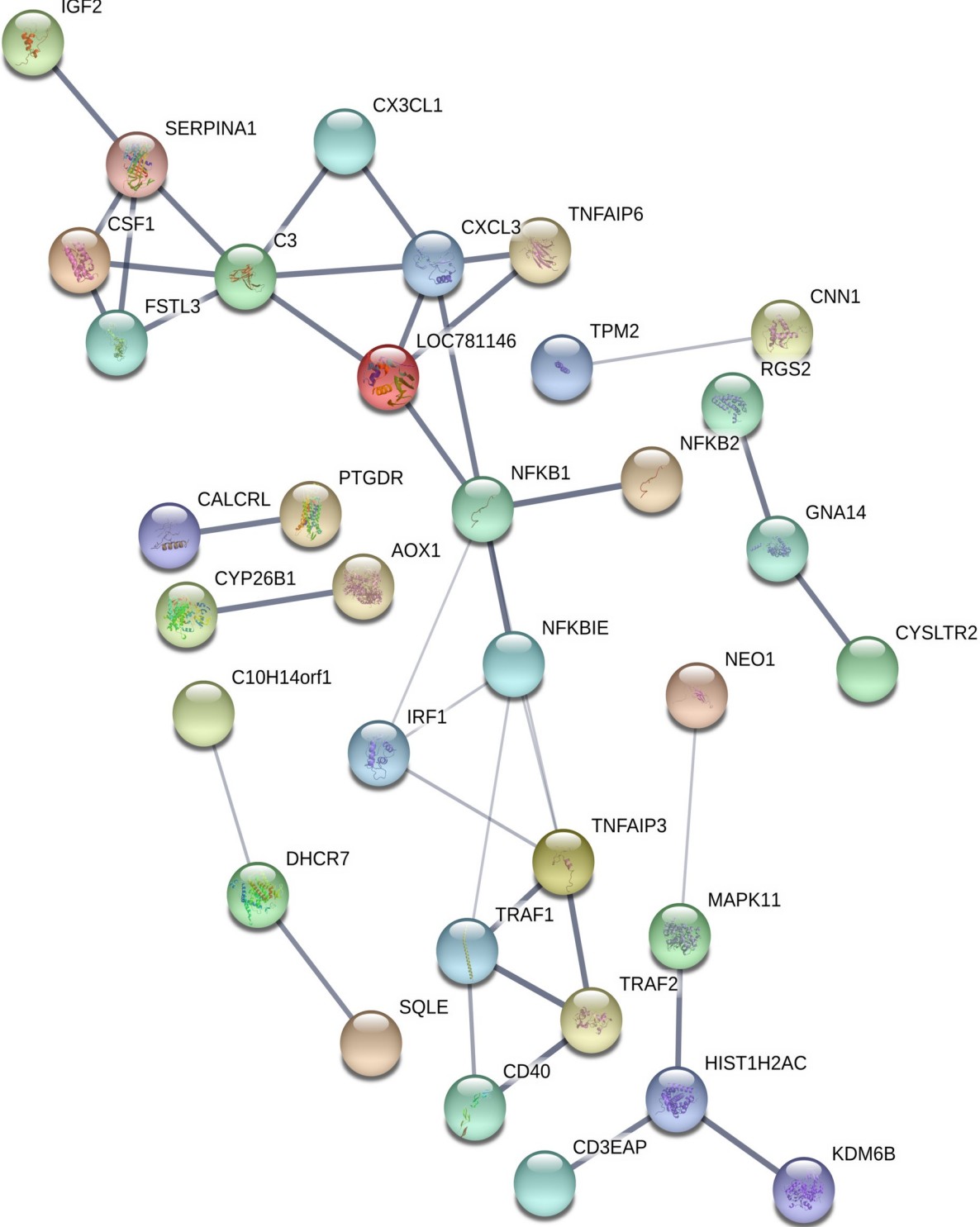

**Fig 4. Predicted protein interaction networks from the DEGs following sPIF treatment.** Interactions are based on the prediction methods of: neighbourhood, gene fusion, co-occurrence, co-expression and experiment databases in STRING version 11.0. Only connected nodes within the DEGs dataset are displayed. Edges between nodes represent predicted protein-to-protein interactions coded by DEGs. Thicker lines demonstrate a greater strength of data support from the prediction methods.

postpartum dairy cattle [52, 53], but heifers and cows exhibit differing endometrial transcriptome responses during early pregnancy [53]. It is for this reason that only heifers were used in the present study to eliminate the effect of previous pregnancies on the data collected. However, as uteri were collected at a local abattoir, heifers were likely from different farms and management backgrounds. Previous studies have demonstrated that nutritional management can also alter endometrial gene expression [54, 55], which could help to explain the variation between animal replicates. Furthermore, the lack of characterisation of the proportions of stromal and epithelial cells within each endometrial explant may also help to explain the strong variation between cattle. Nevertheless, the variation within the data set does not detract from the findings that overall, effects of sPIF on the bovine endometrial transcriptome were relatively small, with no genes being regulated over two-fold.

## Immune signalling

Pathway analysis demonstrated that sPIF plays a coordinated role downregulating genes in the TLR, IL-17, MAPK, TNF and NF-κB signalling pathways, of which the latter two are known to be modulated during the preimplantation period in several species [56–59]. NF-κB signalling is a key component of the TLR, IL-17 and TNF pathways and MAPK signalling is involved in TNF signalling; thus, several of the downregulated genes were common between these key immune related KEGG pathways. Furthermore, from analysis of the DEGs in the TLR and NF-κB KEGG pathway, it became apparent that the TNF receptor superfamily was likely targeted through the downregulation of *CD40*, as well as other intracellular signalling molecules.

Synthetic PIF is recognised to act through a TLR-4 dependent pathway in immune cells [60], but the peptide targets downstream proteins such as thymosin-α1, rather than TLR-4 [61]. The pleiotropic peptide, thymosin-α1 acts on innate immune cells, including CD14 + cells [62, 63], which sPIF is known to target [6, 7] and would likely have been present in the tissue explants in the present study. Modulation of the TLR signalling pathway was largely attributed to DEGs in both the TNF and NF-κB signalling pathways, including downregulation of *CD40*. TLR ligands, including that of TLR-4, modulate *CD40* gene expression in immune cells [64, 65]. Thus, it is hypothesised that if sPIF interacts with cells in bovine endometrial tissue in a TLR-4 dependent manner, *CD40* may link TLR and TNF receptor superfamily signalling. This hypothesis needs further elucidation from functional studies. Furthermore, sPIF induced invasiveness of *in vitro* human extravillous trophoblast cells is blocked through inhibition of the MAPK signalling pathway [66]. As MAPK signalling is also involved in TNF signalling [67], and genes in both pathways were targeted by sPIF in the present study, this adds to evidence that there may be an effect of sPIF on the TNF receptor superfamily signalling pathway.

Downregulation of *CD40* and several of the downstream signalling molecules following sPIF treatment, supports the immune suppressive role of sPIF in bovine endometrium. As a member of the TNF receptor superfamily, *CD40* is involved in inflammatory signalling as part of the adaptive immune response [68–71]. Furthermore, it is suggested that during early pregnancy in mice, increased CD40-CD40L interaction leads to a favouring of the proinflammatory Th1 response, over the predominant Th2 pregnancy response [72]. Thus, this study supports previous research that has identified sPIF to help create a Th2 bias to modulate the maternal uterine immune system, without suppressing the whole system [6, 8].

Excessive exposure to TNF-α has deleterious effects on bovine oocyte development in culture [73] and is associated with pregnancy loss in rat models and human pregnancies [24]. It is suggested that an upregulation of TNFR2 receptors on the bovine endometrium in early pregnancy [59] reduces free TNF protein in uterine fluid [74] which may protect the embryo [59].

Synthetic PIF treatment led to a downregulation of DEGs involved in intracellular signalling following activation of TNF receptors, such as *TRAF1* and *2*, which are recruited to the receptors following ligand binding [68, 69, 75]. Therefore, although there is an increased capacity for ligand binding within the TNF pathway, immune modulators such as PIF, may act on downstream targets to prevent over activation of the maternal TNF pathway and thus, the immune response in early pregnancy.

The NF-κB signalling pathway is activated following TNF and TLR receptor activation [68, 75, 76]. Genes involved in NF-κB signalling are downregulated in early pregnant human decidua [56], mice uteri [57] and porcine endometrium [58] compared to non-pregnant tissue, supported by a downregulation of NF-κB-p65 protein in early pregnancy uterine fluid [74]. Although DEGs following sPIF treatment in the present study were not homologues to those modulated in other species [56–58], the data does support the concept of an immune suppressive state during early bovine pregnancy. Indeed, IFN-τ has also been demonstrated to reduce activation of NF-κB and secretion of proinflammatory cytokines in lipopolysaccharide stimulated RAW264.7 cells [77], suggesting anti-inflammatory actions. Downregulation of the NF-κB signalling pathway in this study therefore suggests a mechanistic explanation for our previous work that demonstrated a reduction in native IL-6 secretion from sPIF treated bovine endometrial explants [27]. Conversely, day 15 bovine conceptuses drive upregulation of the inflammatory response in the endometrium, largely related to TNF and NF-κB signalling [16]. However, sPIF may have a modulatory role within the milieu of conceptus derived factors that act upon the endometrium, preventing overregulation and an imbalance of the inflammatory response, supporting the previously established role of the peptide in promoting a Th2 bias whilst preserving Th1 responses [5, 6, 8, 9].

Genes involved in downstream effects of the NF-κB and TNF signalling pathways were also downregulated in this study following sPIF treatment. Chemokines *CXCL3* and *CX3CL1* [78–80] and adhesion molecules *VCAM1* and *ICAM1* [81, 82], have roles in recruitment and adhesion of leukocytes and are induced in endothelial inflammation [83–86]. The present findings support previous *in vivo* work in mice, whereby sPIF impaired leucocyte recruitment and adhesion in a TNF-α induced inflammatory environment [7]. Furthermore, *in vivo* work has demonstrated that there is a reduction in leukocyte infiltration into the bovine endometrium in early pregnancy [59, 87]. Moreover, *VCAM1* is downregulated in the preimplantation period in pregnant compared to non-pregnant mice uteri, although *ICAM1* is slightly upregulated [57]. Thus, based on the downstream DEGs related with the NF-κB and TNF pathways, it was deemed that sPIF has an immune modulatory effect on the bovine endometrium¸ which supports previous work in cattle [27], horses [88] and humans [5, 49].

The overall immune response to pregnancy is dynamic, whereby the immune tolerant state towards the embryo is also accompanied by some inflammatory responses [24] as protection for the dam, such as increased complement activation [19, 22, 89]. We demonstrated sPIF to upregulate complement component *C3* within the bovine endometrium. *C3* is integral to complement activation, is upregulated in the implantation window in cattle [19] and is suggested to be involved in the maternal to foetal crosstalk around maternal recognition of pregnancy [10]. Furthermore, *LCN2* was upregulated following sPIF treatment. Lipocalin 2 is upregulated around the conceptus fixation site in the endometrium of pregnant mares, with expression likely induced by either the conceptus or its secretory products [90] and also has an innate immune role in the endometrium in response to *E. coli* [91]. Therefore, the present study suggests that sPIF aids protection of the embryo through both immune suppression, to allow the acceptance of the embryo, and also inflammatory responses against invading pathogens.

## Interferon related genes

During early pregnancy in ruminants, the effects of conceptus derived IFN-τ on the maternal endometrium are mediated by the expression of the two receptor subunits, IFNAR1 and IFNAR2, which comprise the type I interferon receptor [12, 92]. Yet in the present study, *IFNAR2* was downregulated following sPIF treatment, which also corresponded with the downregulation of *IRF-1* and *STAT5a*, transcription factors involved in interferon signalling [11, 93]. In contrast to the present study, endometrial expression of *IRF-1* is upregulated by the conceptus in early pregnancy [19, 94] and *IRF-1* and *STAT5a* upregulated by IFN-τ stimulation in the ovine endometrium [93, 95]. PIF is only a small part of the cross talk between the conceptus and endometrium, and therefore other factors are more likely important in the modulation of interferon related genes compared to PIF. Furthermore, the main effects of PIF may be mediated slightly before that of IFN- τ, as at present there are no data to demonstrate the level of secretion of PIF from the elongated filamentous bovine conceptus, compared to earlier developmental stages. However, it must be noted that although involved in IFN-τ signalling, both *IFNAR2* and *IRF-1* were linked to immune networks in the present study (Table 2 and Fig 4), furthermore, *IRF-1* was one of the top 10 most significantly downregulated DEGs. *IRF-1* is also involved in activation of the immune response and apoptosis [96, 97] and has a role in activating genes such as *VCAM-1* [98]. Thus, the downregulation of *IRF-1* in the present study supports the general response of immune related genes and suggests that the downregulation of *VCAM-1* following sPIF treatment could have been controlled by several pathways, further to those described previously.

## Steroid biosynthesis pathway

The steroid biosynthesis pathway was the only upregulated over-represented KEGG pathway following sPIF treatment, with three genes encoding for enzymes, *CYP24A1*, *DHCR7* and *SQLE* being upregulated. These findings are in line with previous work which has shown sPIF to upregulate the expression of genes involved in the cortisol biosynthesis pathway in non-stimulated bovine adrenocortical cells [99]. Furthermore, Binelli *et al.* [21] identified steroid biosynthesis to be an overrepresented pathway in early pregnancy in cattle and also identified *DHCR7* as being upregulated in the pregnant endometrium. Both *DHCR7* and *SQLE* are anabolic enzymes involved in sterol synthesis reactions thus suggesting a need for endometrial anabolic activities in the embryo-maternal crosstalk [21]. *CYP24A1* catalyses the hydroxylation and degradation of calcitriol. Calcitriol has progesterone-like activity in the early stages of gestation in humans, acting on endometrial receptivity and implantation [100]. Circulating concentrations of calcitriol are increased during pregnancy [101] and are suggested to also increase *CYP24A1* expression in a negative feedback system to prevent over activation of the calcitriol system in pregnancy [100]. Thus, in the present study, sPIF had effects on steroid biosynthesis that would be expected in pregnant endometrium.

## Conclusions

In conclusion, sPIF interacts with the bovine endometrium in a manner that suggests that PIF plays a role in early bovine pregnancy. There are some similarities between the mechanisms PIF uses in the bovine endometrium and those defined in the human endometrium, in that sPIF has clear immune modulatory roles to promote tolerance to the embryo, whilst also maintaining the ability to fight invading pathogens. However, the gene expression response to sPIF was much smaller and muted compared to human studies. Further research is now warranted

to better understand the role and, more importantly, the significance of PIF at this critical period of bovine pregnancy.

## Supporting information

**S1 Fig. PCA plot showing all RNA sequencing replicates prior to the two technical replicates for cows 5–7 being summed together.** Variance was evident between the samples on each lane (1 and 2), but not between the technical replicates (2a and 2b) of cows 5–7 which were sequenced twice to ensure similarity in the number of reads between all samples. The first two principle components are displayed.
(PDF)

**S2 Fig. PCA plots demonstrating principle components 1–4.** Variances were detected between animal replicates and samples treated with or without sPIF (100nM). The plot demonstrating principle component 1 and 2 is located in Fig 1B.
(PDF)

**S1 Table. Differentially Expressed Genes (DEG) following sPIF treatment of the bovine endometrium, compared to the control.** Based on P adjusted values ($P$adj<0.1) as assessed by the Bioconductor package, deSeq2 statistical analysis.
(PDF)

**S2 Table. Summary of classes of KEGG pathways significantly over-represented following sPIF treatment.** Based on P adjusted values (False discovery rate: FDR; $P$adj<0.05) as assessed by STRING analysis.
(PDF)

## Acknowledgments

The authors extend their thanks to Dr Eytan Barnea, BioIncept LLC (New Jersey, USA) for the kind donation of sPIF, Dr Colin Sauze for bioinformatics technical support and the staff at Randall Parker Foods for assistance in sample collection.

## Author Contributions

**Conceptualization:** Ruth E. Wonfor, Deborah M. Nash, Michael T. Rose.

**Data curation:** Ruth E. Wonfor, Christopher J. Creevey.

**Formal analysis:** Ruth E. Wonfor, Christopher J. Creevey.

**Funding acquisition:** Deborah M. Nash, Michael T. Rose.

**Investigation:** Ruth E. Wonfor, Manuela Natoli, Matthew Hegarty.

**Methodology:** Ruth E. Wonfor, Christopher J. Creevey, Deborah M. Nash, Michael T. Rose.

**Software:** Christopher J. Creevey.

**Supervision:** Deborah M. Nash, Michael T. Rose.

**Validation:** Ruth E. Wonfor, Deborah M. Nash, Michael T. Rose.

**Visualization:** Ruth E. Wonfor.

**Writing – original draft:** Ruth E. Wonfor.

**Writing – review & editing:** Deborah M. Nash, Michael T. Rose.

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
