## [Decision Letter · Decision Letter 0]

27 May 2020

PONE-D-20-09982

Interaction of preimplantation factor with the global bovine endometrial transcriptome

PLOS ONE

Dear Dr. Wonfor,

Thank you for submitting your manuscript to PLOS ONE. After careful consideration, we feel that it has merit but does not fully meet PLOS ONE’s publication criteria as it currently stands. Therefore, we invite you to submit a revised version of the manuscript that addresses the points raised during the review process.

We look forward to receiving your revised manuscript.

Kind regards,

Juan J Loor

Academic Editor

PLOS ONE

Journal Requirements:

2. We note that you are reporting an analysis of a microarray, next-generation sequencing, or deep sequencing data set. PLOS requires that authors comply with field-specific standards for preparation, recording, and deposition of data in repositories appropriate to their field. Please upload these data to a stable, public repository (such as ArrayExpress, Gene Expression Omnibus (GEO), DNA Data Bank of Japan (DDBJ), NCBI GenBank, NCBI Sequence Read Archive, or EMBL Nucleotide Sequence Database (ENA)). In your revised cover letter, please provide the relevant accession numbers that may be used to access these data. For a full list of recommended repositories, see http://journals.plos.org/plosone/s/data-availability#loc-omics or http://journals.plos.org/plosone/s/data-availability#loc-sequencing

Reviewers' comments:

Reviewer's Responses to Questions

**Comments to the Author**

1. Is the manuscript technically sound, and do the data support the conclusions?

Reviewer #1: Partly

Reviewer #2: Yes

Reviewer #3: Partly

2. Has the statistical analysis been performed appropriately and rigorously? 

Reviewer #1: N/A

Reviewer #2: Yes

Reviewer #3: Yes

3. Have the authors made all data underlying the findings in their manuscript fully available?

Reviewer #1: Yes

Reviewer #2: Yes

Reviewer #3: No

4. Is the manuscript presented in an intelligible fashion and written in standard English?

Reviewer #1: Yes

Reviewer #2: Yes

Reviewer #3: Yes

5. Review Comments to the Author

Reviewer #1: This manuscript describes the effect of sPIF on the cow endometrium, confirming its role in modulating the immune balance between acceptance and rejection of the conceptus at the beginning of pregnancy. This represents overall an interesting piece of information which would deserve publication. However, major points related to the methodology used resulting mainly from weaknesses in the characterization of the biological samples should be addressed. Additional work to better characterize the samples would be the source of valuable improvements. This would allow revisiting the statistical analysis by introducing some pertinent co-variables which may give more strength to the results. At present, the results of the differential gene expression analysis (although consistent with those of previous studies) are not fully demonstrative due to low significance of DEGs identified, related probably to the existence of “background noise” generated by the heterogeneity of the samples. This situation creates a lack of power. The way results are affected by the above defaults in methodology should be discussed. Differences in constitutive gene expression related to individual biological samples and how these differences influence response to sPIF should be addressed in a more complete way.

Overall, the discussion is very long and some parts redundant. Although central in the discussion, the part on “ Immune signaling” is extremely long and should probably be shortened.

Other comments:

Lines 43-44: “whilst preventing suppression of the whole immune response”… this concept is not fully clear, looks complex at this stage of the reading and one may question what is the real meaning of this part of the sentence. Things are well explained later on lines 47-51 and then it is easy to understand, but sentences in between makes the link less obvious…

“whilst preventing suppression of the whole immune response” could be suppressed in this sentence, line 44 and then placed later before the detailed explanation about immune mechanisms is given. ….

Lines 45-46: The sentence is somewhat ambiguous. It would probably more clear if the authors refer to four pathways as adhesion and apoptosis or apoptosis and tissue remodeling also could be seen as two different ones.

Line 48: Targets CD14+ monocular cells and then do what ?

Lines 58-59: “conception rates” should be preferred instead of “reproductive rates” which is really vague ….

Line 60: “Several studies have attempted to understand the bovine….” could be replaced by “Several attempts aimed at understanding the bovine….”

Lines 74-75: “Due to differences in the maternal recognition of pregnancy in humans compared to cattle, it was deemed likely that the role of PIF will be different between these species”. As it is well explained above that the role of PIF relates essentially to immune mechanisms (immunosuppression / tolerance and preservation of other types of immune reaction) which are potentially common mechanisms existing in the two species, this sentence looks somewhat confusing and does not bring anything to clarify the text at this stage of lecture. Due to results of the present study, it is OK to mention similarities in reactions to sPIF in human and bovine as mentioned in the conclusion lines 485-489.

Line 97: Sentence should be replaced by “The limit of detection of the progesterone assay was …”

Line 100: Sentence should be replaced by “ … using the method described by Borges et al., (34).

Lines 101-102: Some important information is lacking in the description. The place where punches were made was chosen at random ? or systematically performed at a given place / for instance distance from UTJ. More importantly, as gene expression /overall transcriptome is potentially submitted to very important variations due to the respective amounts of stromal and epithelial cells of the samples, it should be mentioned if explants were taken from caruncular or inter-caruncular tissue. Several punches were performed per uterine horn/cow ?

A major flaw from the present study is the lack of (description of?) characterization of the samples. The respective proportions of stromal and epithelial cells for each of the tissue samples should be determined to see if differences between samples can explain such a variability allowing later on adjustments of the RNAseq results. This should be done if possible by additional work from remaining parts of samples.

Line 135: “… then samples were pooled…” as mentioned above the number of samples and their origin , is not clear. It is said later lines 139-140 that 14 samples were sequenced meaning that 2 explants per cow could have been taken … but in that case what is the meaning of “samples were pooled” ?

Lines 163-164: It is clear that samples are paired and should be treated this way. However, again, it is not clear if the treated and control sample originates from the same biopsy/explant cut into two pieces (exposed or not to PIF) or from two different ones which is less good due to comments lines 101-102 ….

Lines 167-168: The progesterone concentrations especially in the group > 1ng/ml should be more documented (at least the range should be given) to illustrate the variation in this group and especially to show the existence of any “outlier” (and their number) with relatively high progesterone concentrations. It is shown in the result section that there is 4 cows with progesterone concentrations >1ng/ml. A mean of 3.1 +/- 0.86 (is it SD or SEM ?) means that some samples were around 5… these should be identified and located in the PCA. It means also that some of the cows were probably close to the cut-off chosen. Due to this it could have been better to use progesterone as a co-variable in the model instead of making two classes. The statistical analysis for differential gene expression should be revisited that way.

Lines 169-170 and later on in the result section : The p adjusted value of 0.1 is not classical…What will be the number of DEGs at the conventional level of p<0.05 ?

Lines 189-180: This sentence refers to 7 samples analyzed by RNAseq whereas 14 are mentioned above lines 139-140. I was thinking analyses were based on 7 controls and 7 treated by PIF samples, then I am lost. These relates also to earlier comments about the identification of samples analyzed (lines 135 and 163-164). This point is really confusing. Then looking at the figures it is clear that 14 samples were analyzed…

Lines 194-197: Table 1 and S1 are not commented at all.

Lines 200-220, Table 1: It should be preferable to use “over-expressed” and “under-expressed” than “up-regulated” and “down-regulated” because at this stage results are simply descriptive and do not provide evidence for a regulatory role of PIF on all these genes. Due to the fact that cut off was placed at padj<0.10, adjusted p values should also be presented to see if some were close to p<0.05.

Lines 222-228: This part should probably take place before the analysis of the effects of PIF. Looking at the PCA results, it appears that the “overall” effect of treatment is really cow dependent typical of an interaction which could not be tested here.

Lines 230-234: Sometimes other dimensions reveal better possible differences. Was this approach tested ?

Line 237 , line 242: Would be better to use “DEGs” instead of “DEG” as the ontology group or pathway includes several genes …. Same in all text when appropriate …

Lines 310-318: This part of the discussion should be revisited to take into account some of the weaknesses of the methods used. The fold change reported in the human species refers to specific populations of cells, whereas the results obtained here are issued from full tissue consisting of different types of cells. The strong variation observed between animals and also in the way PIF affects overall expression reflected by Fig 1 is probably the result of analyses performed from full biopsies which is source of heterogeneity as stromal and epithelial cells could express different types of responses… (see comments lines 100-102).The discussion should at least be modified to indicate that the changes observed here in response to sPIF are very limited (few number of genes, with low fold change …) but probably true, as this lack of characterization is source of background noise and low significance.

Lines 438-473: In relation with the above point, taking in consideration the factors mentioned in the analysis was OK but could not compensate the impact of other more important sources of variation. This point could be discussed as well. In general, the methodological issues should be discussed at first. Then considering the limitations induced by these the discussion about impact of sPIF could follow.

Lines 319-327: The way things are expressed here is somewhat redundant. This part could be shortened and the information presented in a more synthetic way.

Line 333: Senyence should be better replaced by “ Furthermmore, from analysis of the genes …”

Line 343: Redundant with lines 341-343.

Line 349: Could be replaced by “further elucidation from functional studies.”

Line 369: Could be replaced by “… interface which may protect the embryo.”

Reviewer #2: This paper describes the transcriptome response to treating bovine endometrial explants with synthetic human Preimplantation Factor (PIF). The study has been performed and analysed in an appropriate manner. The results are also presented and discussed clearly.

Comments are as follows.

1. General: it would be helpful in the Introduction and/or the discussion to describe the timelines for relative production of PIF and IFNT in more detail. The discussion implies that there are some contradictions in terms of their actions on local immunity in the endometrium, but most work on PIF has been performed at an earlier stage of pregnancy than the time when IFNT is produced.

2. Abstract Line 17. Suggest removing the word “novel”. PIF has been known about for quite a long time now.

3. Line 102. More detail of the culture method is needed. It is said that the punches were weighed, but not how much they each weighed or the total weight placed into each well. I am unclear as to how many punches were used per well. This also relates to Line 114 where it is stated that explants were stored individually and line 121 which says that <20 mg of frozen tissue was extracted.

4. Line 113. No explanation is provided as to why a 30h culture period was chosen, or why the medium change took place after 24h.

5. Line 248. I commend the authors for discarding the irrelevant pathways – many people don’t!

6. L325. I am unsure that the reference to the “quiet embryo” hypothesis is relevant when talking about down-regulation of uterine immunity. This paper was looking at the metabolism of the embryo between fertilization and early blastocycst, when is mostly located in the oviduct. Again this comes back to understanding the timeline and what is meant by “early” with respect to the actions of PIF. There are many important changes in the endometrial transcriptome during the relatively long pre-implantation period in the cow, initially controlled by the timing of the progesterone rise before production of IFNT begins.

Reviewer #3: This paper describes a study whereby endometrial explants were cultured in vitro. One set of samples was treated with synthetic preimplantation factor peptide. Following a 30h in vitro culture of the explants the authors carried out a transcriptome study by RNAseq. There were 102 genes inferred as differentially expressed between tissues that were treated with synthetic preimplantation factor peptide relative to the control. The authors followed up with functional characterization of the genes using in silico approaches. Overall the paper is well organized, but need improvements prior to receiving further consideration for publication.

Statement of data availability: “Yes - all data are fully available without restriction”

This is definitely not the case

“Data files will be available from NCBI Gene Expression Omnibus database.”

Well, IF that is the case, let the reviewers see it.

Lines 100-106: There should be more details on the sampling of endometrium. For instance, how thick was the explant? Were intercaruncular and caruncular regions sampled?

Lines 171-174: Please, specify the list of genes used on the background for the calculation of gene enrichment on these 3 databases.

Line 174: “significance set at P<0.05.”

Why there was not an adjustment for multiple hypothesis testing?

Line 178: “however due to the discovery of several false positive results”

How did you access false positive results?

How can you assure that there were no longer false positive results on the other prediction methods (neighbourhood, gene fusion, co-occurrence, co-expression)?

Lane 191: “15,682 transcripts were analysed for differential expression”

This implies to me that there was a filtering of genes. Please detail the criteria for this filtering in the methods.

Table 1. Authors need to adjust the scientific notation for numbers on the FDR column. The coefficient is written between 1 and 10.

See https://en.wikipedia.org/wiki/Scientific_notation

Figure 1b is not referenced in the text.

To me it would be more logical to start the results with the information on the general characterization of the data and samples, which is presented on fig 1, then follow with identification of DEGs based on the sPIF treatment.

Lines 237-240: Would it not be important to show what genes were present in these categories?

Table 2. I do not understand the value of table 2. Perhaps it can be moved to supplementary materials. The authors already placed relevant KEGG pathways on table 3.

The legend in Figures 2 and 3 need to be changed to “Predicted changes in the ….” or “Putative changes in the ….”

Lane 296: “(p=2.71 x108) with a total of 30”

What was the gene list used for the background considered for the calculation of this p value?

Lane: 306: “Stronger associations are”

How was association’s strength assessed?

Lines 476:484: This paragraph is relevant, but should be moved to the discussion section.

6. PLOS authors have the option to publish the peer review history of their article (what does this mean?). If published, this will include your full peer review and any attached files.

Reviewer #1: No

Reviewer #2: No

Reviewer #3: No

---

## [Author Response · Author response to Decision Letter 0]

9 Jul 2020

The authors would like to express our sincere thanks to the editor and the reviewers for taking the time to read our manuscript and for helping us to make it better – your work is very much appreciated. Our full response is available in the 'response to reviewers' document in a clear table, and outlined below. We have indicated the new lines where you can find the revisions on the highlighted revised manuscript with track changes.

Journal Requirements

Please ensure that your manuscript meets PLOS ONE's style requirements, including those for file naming: Formatting has been checked and amended in places such as author list. We have named files as requested in the editor’s response to authors.

We note that you are reporting an analysis of a microarray, next-generation sequencing, or deep sequencing data set. PLOS requires that authors comply with field-specific standards for preparation, recording, and deposition of data in repositories appropriate to their field. Please upload these data to a stable, public repository (such as ArrayExpress, Gene Expression Omnibus (GEO), DNA Data Bank of Japan (DDBJ), NCBI GenBank, NCBI Sequence Read Archive, or EMBL Nucleotide Sequence Database (ENA)). In your revised cover letter, please provide the relevant accession numbers that may be used to access these data: The data has been uploaded to GEO under the series record GSE153699. Individual samples accession numbers are as follows:

GSM4649298 Bovine endometrium_1_Control 

GSM4649299 Bovine endometrium_1_sPIF 

GSM4649300 Bovine endometrium_2_Control 

GSM4649301 Bovine endometrium_2_sPIF 

GSM4649302 Bovine endometrium_3_Control 

GSM4649303 Bovine endometrium_3_sPIF 

GSM4649304 Bovine endometrium_4_Control 

GSM4649305 Bovine endometrium_4_sPIF 

GSM4649306 Bovine endometrium_5_Control 

GSM4649307 Bovine endometrium_5_sPIF 

GSM4649308 Bovine endometrium_6_Control 

GSM4649309 Bovine endometrium_6_sPIF 

GSM4649310 Bovine endometrium_7_Control 

GSM4649311 Bovine endometrium_7_sPIF 

The record is currently set to private before publications, but can be reviewed at: https://www.ncbi.nlm.nih.gov/geo/query/acc.cgi?acc=GSE153699, with the reviewer token: ulslcyyonvivtgd 

We have clarified sequencing of samples from animals 5-7 which were sequenced twice due to a sample loading error which resulted in low reads compared to animals 1-4 (Lines 158-161). We have also included an assessment of the mean counts between the two sample groups on different lanes (Lines 224-231) and highlighted that the technical replicates for each sample clustered together on a PCA plot, demonstrating that they were good technical replicates and therefore appropriate to be included together in the analysis (Lines 229-231; S1 FIg). 

Please include captions for your Supporting Information files at the end of your manuscript, and update any in-text citations to match accordingly: Lines 916-930: Captions for supporting information now added

Reviewer #1:

However, major points related to the methodology used resulting mainly from weaknesses in the characterization of the biological samples should be addressed. Additional work to better characterize the samples would be the source of valuable improvements. This would allow revisiting the statistical analysis by introducing some pertinent co-variables which may give more strength to the results. At present, the results of the differential gene expression analysis (although consistent with those of previous studies) are not fully demonstrative due to low significance of DEGs identified, related probably to the existence of “background noise” generated by the heterogeneity of the samples. This situation creates a lack of power. The way results are affected by the above defaults in methodology should be discussed. Differences in constitutive gene expression related to individual biological samples and how these differences influence response to sPIF should be addressed in a more complete way: We understand the reviewer’s concerns over weaknesses in the characterisation of the biological samples. In response to these comments we have made some further clarifications which we feel have strengthened the manuscripts clarity with regards to the samples and background noise generated by the samples. Firstly, we have given further information in relation to the mean counts data for each gene and highlighted where the majority of our DEGs sit within the range of the count data (Lines 220-231). Secondly, we have clarified how the progesterone grouping was included in the analysis as an interaction effect in the design (Lines 190-193). Finally, we have given greater discussion to the limitations of the explant model in relation to the characterisation of the epithelial and stromal cell content per explant (Lines 386-397). 

Overall, the discussion is very long and some parts redundant. Although central in the discussion, the part on “ Immune signaling” is extremely long and should probably be shortened: We have shortened the discussion substantially, especially for the immune response section, while incorporating the reviewers’ other requirements.

Lines 43-44: “whilst preventing suppression of the whole immune response”… this concept is not fully clear, looks complex at this stage of the reading and one may question what is the real meaning of this part of the sentence. Things are well explained later on lines 47-51 and then it is easy to understand, but sentences in between makes the link less obvious…“whilst preventing suppression of the whole immune response” could be suppressed in this sentence, line 44 and then placed later before the detailed explanation about immune mechanisms is given: Line 47: We have removed the statement “…whilst preventing suppression of the whole immune response” to avoid confusion in this sentence and left the remaining subsequent explanation to provide clarity. 

Lines 45-46: The sentence is somewhat ambiguous. It would probably more clear if the authors refer to four pathways as adhesion and apoptosis or apoptosis and tissue remodeling also could be seen as two different ones: Line 49: apoptosis and remodelling of the uterus are meant to be classed as one pathway, based on the analysis by Paidas et al. (2010). Therefore, we have removed the ‘and’ between these statements and added “apoptosis/remodelling of the uterus”

Line 48: Targets CD14+ monocular cells and then do what ?: Line 52: We have clarified what the action of sPIF on naïve CD14+ PBMCs is and added “and reduces secretion and mRNA expression of Th1/Th2 cytokines”.

Lines 58-59: “conception rates” should be preferred instead of “reproductive rates” which is really vague ….: Line 65: We agree that reproductive rates is a vague term however, conception rates does not correctly reflect that we are describing maintenance of pregnancy, rather than establishment of a pregnancy. Therefore, we have amended the text to: “fertility rates”.

Line 60: “Several studies have attempted to understand the bovine….” could be replaced by “Several attempts aimed at understanding the bovine….”: Line 60: We have amended the text to: “Several attempts have aimed at understanding the bovine…”

Lines 74-75: “Due to differences in the maternal recognition of pregnancy in humans compared to cattle, it was deemed likely that the role of PIF will be different between these species”. As it is well explained above that the role of PIF relates essentially to immune mechanisms (immunosuppression / tolerance and preservation of other types of immune reaction) which are potentially common mechanisms existing in the two species, this sentence looks somewhat confusing and does not bring anything to clarify the text at this stage of lecture. Due to results of the present study, it is OK to mention similarities in reactions to sPIF in human and bovine as mentioned in the conclusion lines 485-489.: Lines 79-83: We have clarified that there are likely to be similarities between the human and bovine but highlighted that as there are differences around early pregnancies, there are also likely to be some difference in the way PIF acts between species. The text has been amended as follows: “Synthetic PIF is hypothesised to have an immune modulatory role in cattle, similar to that described in the human. Although, due to differences in the maternal recognition and early pregnancy in humans compared to cattle, it was deemed likely that there would be some differences in the role of PIF between these species.”

Line 97: Sentence should be replaced by “The limit of detection of the progesterone assay was …”: Lines 104-105: This has been amended in the text

Line 100: Sentence should be replaced by “ … using the method described by Borges et al., (34).: Line 107: This has been amended in the text

Lines 101-102: Some important information is lacking in the description. The place where punches were made was chosen at random ? or systematically performed at a given place / for instance distance from UTJ. More importantly, as gene expression /overall transcriptome is potentially submitted to very important variations due to the respective amounts of stromal and epithelial cells of the samples, it should be mentioned if explants were taken from caruncular or inter-caruncular tissue. Several punches were performed per uterine horn/cow ?

A major flaw from the present study is the lack of (description of?) characterization of the samples. The respective proportions of stromal and epithelial cells for each of the tissue samples should be determined to see if differences between samples can explain such a variability allowing later on adjustments of the RNAseq results. This should be done if possible by additional work from remaining parts of samples.: We have made considerable improvements to the clarity of the Materials and Methods section, but primarily in the Endometrial Explant tissue culture section – Lines 107-127. We have clarified that we:

• “sampled randomly from the intercaruncular tissue in the first third (closest to the utero-tubular junction) of the uterine horn ipsilateral to the staged ovary.” 

• Collected a total of six biopsies per animal, but that RNA was only extracted from 2 explants (one for each treatment, control or sPIF) per animal.

We recognise the reviewer’s concerns over the potential variability arising from the proportions of epithelial and stromal cells in each explant sample. However, we are unable to complete further analysis of the cell content in explants due to age of samples, a lack of remaining funding for this project and time restraints due to having no lab access at present with current COVID-19 restrictions. To mitigate the lack of analysis we have added in a discussion of this point and the implications the potential variable cell content in each explant may have had on the analysis, as well as a justification as to our reasoning behind choosing this method instead of individual cell types (Lines 386-397). We also note that a recent study (Mathew et al. (2019) Biol Reprod. 100(2):365-80. doi: 10.1093/biolre/ioy199) uses the same method as us to assess the effect of bovine conceptuses and IFN-τ on the endometrial transcriptome, without characterising the populations of epithelial and stromal cells within each sample and we have added this reference to our discussion on the methodology. 

Line 135: “… then samples were pooled…” as mentioned above the number of samples and their origin , is not clear. It is said later lines 139-140 that 14 samples were sequenced meaning that 2 explants per cow could have been taken … but in that case what is the meaning of “samples were pooled”?: Pooling of the samples at this point in the RNAseq library preparation is after unique-barcoding of the samples and is in relation to the library preparation and sequencing process. We have clarified this in the text (Lines 151-152).

Lines 163-164: It is clear that samples are paired and should be treated this way. However, again, it is not clear if the treated and control sample originates from the same biopsy/explant cut into two pieces (exposed or not to PIF) or from two different ones which is less good due to comments lines 101-102 ….:Lines 121-122: We have clarified that: “Whole explant biopsies from each animal were treated with either medium alone or with sPIF (100nM) for 24 h in 6 well plates.”

Lines 167-168: The progesterone concentrations especially in the group > 1ng/ml should be more documented (at least the range should be given) to illustrate the variation in this group and especially to show the existence of any “outlier” (and their number) with relatively high progesterone concentrations. It is shown in the result section that there is 4 cows with progesterone concentrations >1ng/ml. A mean of 3.1 +/- 0.86 (is it SD or SEM ?) means that some samples were around 5… these should be identified and located in the PCA. It means also that some of the cows were probably close to the cut-off chosen. Due to this it could have been better to use progesterone as a co-variable in the model instead of making two classes. The statistical analysis for differential gene expression should be revisited that way.: Lines 210-213: We have now provided the range to reflect the reviewer’s comments that the variation in the group needed to be demonstrated. We have deemed that there were no outliers in the >1ng/ml (Values were: 1.46; 1.44; 4.1 and 5.41 ng/mL). Furthermore, on the PCA plots (Fig 1 and S2 Fig), there is no clear definition between samples with differing progesterone concentrations that suggests there is a clear grouping of the progesterone concentration to be labelled. 

We have also clarified that the ± value provided was the SEM. We have used the progesterone grouping based on previous work (Wonfor et al., 2017, doi: 10.1016/j.theriogenology.2017.08.001; Saut et al., 2014, DOI: 10.1530/REP-14-0230) that has split uterine tissue samples based on the stage of cycle (Stage IV ovaries and progesterone concentration <1ng/ml). Thus, although all uteri were deemed to have a stage IV ovary by visual examination, we have controlled for potential effects of high progesterone concentrations by splitting our data into these two groups and making our current methodology comparable to the way we have handled data previously. 

Lines 169-170 and later on in the result section : The p adjusted value of 0.1 is not classical…What will be the number of DEGs at the conventional level of p<0.05 ? A Padj value of <0.1 is commonly used in RNA-seq studies. Examples of references that we have based our use of Padj<0.1 are as follows:

• Binelli et al. (2015) PLoS One. 10(4):e0122874. doi: 10.1371/journal.pone.0122874.

• Mathew et al. (2019) Biol Reprod. 100(2):365-80. doi: 10.1093/biolre/ioy199.

• McCabe et al. (2012) BMC Genomics. 13:193. doi.org/10.1186/1471-2164-13-193

• Moran et al. (2017) Reprod, Fert and Dev. 29(2): 274-282 DOI: 10.1071/RD15128

Lines 189-180: This sentence refers to 7 samples analyzed by RNAseq whereas 14 are mentioned above lines 139-140. I was thinking analyses were based on 7 controls and 7 treated by PIF samples, then I am lost. These relates also to earlier comments about the identification of samples analyzed (lines 135 and 163-164). This point is really confusing. Then looking at the figures it is clear that 14 samples were analyzed…: Line 218: We have amended the text to 14 samples.

Lines 194-197: Table 1 and S1 are not commented at all.:Table 1 is referred to on line 256 and we have added a further comment on the table on lines 256-258. S1 Table is referred to on line 256, this is provided for the readers’ reference.

Lines 200-220, Table 1: It should be preferable to use “over-expressed” and “under-expressed” than “up-regulated” and “down-regulated” because at this stage results are simply descriptive and do not provide evidence for a regulatory role of PIF on all these genes. Due to the fact that cut off was placed at padj<0.10, adjusted p values should also be presented to see if some were close to p<0.05.:Lines 264-283: Text in Table 1 has been amended to “over-expressed” and “under-expressed” from “up-regulated” and “down-regulated”. Padj values are provided in the FDR column of Table 1 to demonstrate that all of the top 10 DEG were padj<0.05

Lines 222-228: This part should probably take place before the analysis of the effects of PIF. Looking at the PCA results, it appears that the “overall” effect of treatment is really cow dependent typical of an interaction which could not be tested here.: We have now amended the structure of the Results section to reflect these comments. We added a subheading “RNA-sequencing overview” to house the general information on the sequencing, then moved the “Sample variability” subheading before “Identification of differentially expressed genes”.

Lines 230-234: Sometimes other dimensions reveal better possible differences. Was this approach tested ?: We have rerun the PCA with a different R package which completes a more thorough analysis than that offered through deSeq2. We have therefore replaced the PCA in Fig 1b with the new PCA assessing PC1 and PC2. Furthermore, we assessed the explained variation in each PC and assessed through the Elbow method and Horn’s parallel analysis that the optimum number of PCs to retain were the first 4. These PCA plots are now displayed in S2 Fig. Although there are no clear further differences to comment on, we have provided a commentary that PC1 accounts for the variation related to the different lanes (Lines 237-239), and that there was no clear clustering of the high or low progesterone groups (Lines 241-244). 

Line 237 , line 242: Would be better to use “DEGs” instead of “DEG” as the ontology group or pathway includes several genes …. Same in all text when appropriate …:We have amended DEG to DEGs throughout the text in the whole manuscript 

Lines 310-318: This part of the discussion should be revisited to take into account some of the weaknesses of the methods used. The fold change reported in the human species refers to specific populations of cells, whereas the results obtained here are issued from full tissue consisting of different types of cells. The strong variation observed between animals and also in the way PIF affects overall expression reflected by Fig 1 is probably the result of analyses performed from full biopsies which is source of heterogeneity as stromal and epithelial cells could express different types of responses… (see comments lines 100-102).The discussion should at least be modified to indicate that the changes observed here in response to sPIF are very limited (few number of genes, with low fold change …) but probably true, as this lack of characterization is source of background noise and low significance. :Lines 386-397 & 420-422: As stated in previous comments, we have added a paragraph that discusses the methodology within this paper and highlights that we have assessed the effect of sPIF on the bovine endometrium in a tissue explant model, rather than on individual cell types, which warrants further study. We have also acknowledged that this difference in methodologies between our study and previous work in humans in relation to sPIF, may explain the weaker response to sPIF in our study. 

Lines 438-473: In relation with the above point, taking in consideration the factors mentioned in the analysis was OK but could not compensate the impact of other more important sources of variation. This point could be discussed as well. In general, the methodological issues should be discussed at first. Then considering the limitations induced by these the discussion about impact of sPIF could follow.: In response to this comment we have restructured the discussion so that after our initial summary of the study, we then discuss the limitations of the work, followed by the variation between animal replicates (now moved to lines 407-424). In the variability subsection, we have also added a sentence that again highlights that some of the variation between cattle may stem from the lack of characterisation of the stromal and epithelial cell content of each sample (Line 420-422). 

Lines 319-327: The way things are expressed here is somewhat redundant. This part could be shortened and the information presented in a more synthetic way.:In response to reviewer 2 we have removed this paragraph (now located at lines426-434), apart from the first sentence which has been moved to line 379-380.

Line 333: Senyence should be better replaced by “ Furthermmore, from analysis of the genes …”: Line 440: We have amended the text as suggested. 

Line 343: Redundant with lines 341-343.: Lines 449-452: We have amended the text the be more succinct to “Modulation of the TLR signalling pathway was largely attributed to DEGs in both the TNF and NF-κB signalling pathways, including downregulation of CD40.”

Line 349: Could be replaced by “further elucidation from functional studies.”: Line 457: We have amended this in the text

Line 369: Could be replaced by “… interface which may protect the embryo.”: Lines 481: As part of the shortening of the discussion, some of this sentence has been removed, however, we have added “which may protect the embryo”, as requested.

Reviewer #2:

1. General: it would be helpful in the Introduction and/or the discussion to describe the timelines for relative production of PIF and IFNT in more detail. The discussion implies that there are some contradictions in terms of their actions on local immunity in the endometrium, but most work on PIF has been performed at an earlier stage of pregnancy than the time when IFNT is produced.: We have now clarified the timelines for PIF and IFNT production in more detail in the introduction. Further detail on PIF can be found on lines 41-44 and IFNT can be found on lines 56-60. 

We have also added a statement into lines 553-555of the discussion to reflect the implication of contradictions on local immunity. 

2. Abstract Line 17. Suggest removing the word “novel”. PIF has been known about for quite a long time now.: Line 17: We have removed the word “novel”. 

3. Line 102. More detail of the culture method is needed. It is said that the punches were weighed, but not how much they each weighed or the total weight placed into each well. I am unclear as to how many punches were used per well. This also relates to Line 114 where it is stated that explants were stored individually and line 121 which says that <20 mg of frozen tissue was extracted. :We have made considerable improvements to the clarity of the Materials and Methods section, but primarily in the Endometrial Explant tissue culture section – Lines 106-127. We have clarified:

• The mean ± SD weight of the explants (Lines 111-112)

• That one biopsy was placed per well of a 6 well plate (Line 112)

• That the extraction of RNA was completed from two explants (one for each treatment, control or sPIF) per animal, and that from these explants, <20mg was removed for the extraction process (Lines 129-134).

4. Line 113. No explanation is provided as to why a 30h culture period was chosen, or why the medium change took place after 24h.: Line 123: Our choice of timing was based on a previous methodology in Wonfor et al. (2017). We have added this into the manuscript for clarification. 

5. Line 248. I commend the authors for discarding the irrelevant pathways – many people don’t!: We thank the reviewer for your commendation. 

6. L325. I am unsure that the reference to the “quiet embryo” hypothesis is relevant when talking about down-regulation of uterine immunity. This paper was looking at the metabolism of the embryo between fertilization and early blastocycst, when is mostly located in the oviduct. Again this comes back to understanding the timeline and what is meant by “early” with respect to the actions of PIF. There are many important changes in the endometrial transcriptome during the relatively long pre-implantation period in the cow, initially controlled by the timing of the progesterone rise before production of IFNT begins. :A useful point to be made. Combined with a comment from reviewer 1, we have now removed this paragraph which references the “quiet embryo”. 

Reviewer #3:

Statement of data availability: “Yes - all data are fully available without restriction”

This is definitely not the case

“Data files will be available from NCBI Gene Expression Omnibus database.”

Well, IF that is the case, let the reviewers see it.: Data are available on GEO. Accession numbers can be found in the comment on Journal requirements above. 

Lines 100-106: There should be more details on the sampling of endometrium. For instance, how thick was the explant? Were intercaruncular and caruncular regions sampled?:

We have made considerable improvements to the clarity of the Materials and Methods section, but primarily in the Endometrial Explant tissue culture section – Lines 106-127. We have clarified that :

• Lines 108-110: We “sampled randomly from the intercaruncular tissue in the first third (closest to the utero-tubular junction) of the uterine horn ipsilateral to the staged ovary.” 

• Lines 110-111: “The endometrial tissue was then dissected away from the myometrium using sterile scissors.”

We did not measure the thickness of the explant, but we have clarified that the endometrial tissue was dissected away from the myometrium. We have also provided the mean ± SD weight of the explants (Lines 111-112).

Lines 171-174: Please, specify the list of genes used on the background for the calculation of gene enrichment on these 3 databases.: Line 197: we have clarified that the background used for these calculations was the Bos taurus genome.

Line 174: “significance set at P<0.05.” Why there was not an adjustment for multiple hypothesis testing?: Line 199: We apologise for the mistake, this is now corrected to Padj<0.05.

Line 178: “however due to the discovery of several false positive results” How did you access false positive results? How can you assure that there were no longer false positive results on the other prediction methods (neighbourhood, gene fusion, co-occurrence, co-expression)?: Lines 204-205: We have amended the term “false-positive” which was misleading and changed to “non-specific” as well as stating that this demonstrated a more focussed network with the removal of text mining. 

Lane 191: “15,682 transcripts were analysed for differential expression”This implies to me that there was a filtering of genes. Please detail the criteria for this filtering in the methods. :Lines 183-184: We have clarified that deSeq2 removes an genes from the statistical model that have less than 10 counts for any one sample. 

Table 1. Authors need to adjust the scientific notation for numbers on the FDR column. The coefficient is written between 1 and 10. :Table 1: The scientific notation for FDR has been amended in the table

Figure 1b is not referenced in the text.: Lines 235 and 239: Figure 1b is referenced here 

To me it would be more logical to start the results with the information on the general characterization of the data and samples, which is presented on fig 1, then follow with identification of DEGs based on the sPIF treatment.: We have amended this also in response to comments from reviewer 1: ‘We have now amended the structure of the Results section to reflect these comments. We added a subheading “RNA-sequencing overview” to house the general information on the sequencing, then moved the “Sample variability” subheading before “Identification of differentially expressed genes”.’

Lines 237-240: Would it not be important to show what genes were present in these categories?: Table 2 now demonstrated genes present in each of the 2 GO categories that were over-represented. 

Table 2. I do not understand the value of table 2. Perhaps it can be moved to supplementary materials. The authors already placed relevant KEGG pathways on table 3.: The original Table 2 has now been moved to the supplementary materials S2 Table, as requested. 

The legend in Figures 2 and 3 need to be changed to “Predicted changes in the ….” or “Putative changes in the ….” We have now added “Putative changes” to the figure legends of Fig 2 and 3. 

Lane 296: “(p=2.71 x108) with a total of 30”What was the gene list used for the background considered for the calculation of this p value?: Line 361: we have now clarified here and in the Materials and Methods that the background used for these calculations was the Bos taurus genome.

Lane: 306: “Stronger associations are”How was association’s strength assessed?: Line 372-373: We have amended the term “Stronger associations” to demonstrate that the “Thicker lines demonstrate a greater strength of data support from the prediction methods”

Lines 476:484: This paragraph is relevant, but should be moved to the discussion section.: Lines 398-406: We have now moved this paragraph towards the beginning of the discussion where we feel that it fits more appropriately with the discussion on some of the weaknesses of the study.

---

## [Decision Letter · Decision Letter 1]

19 Aug 2020

PONE-D-20-09982R1

Interaction of preimplantation factor with the global bovine endometrial transcriptome

PLOS ONE

Dear Dr. Wonfor,

Thank you for submitting your manuscript to PLOS ONE. After careful consideration, we feel that it has merit but does not fully meet PLOS ONE’s publication criteria as it currently stands. Therefore, we invite you to submit a revised version of the manuscript that addresses the points raised during the review process.

PLEASE ADDRESS CAREFULLY ISSUES RAISED BY REVIEWER #3.

We look forward to receiving your revised manuscript.

Kind regards,

Juan J Loor

Academic Editor

PLOS ONE

Reviewers' comments:

Reviewer's Responses to Questions

**Comments to the Author**

1. If the authors have adequately addressed your comments raised in a previous round of review and you feel that this manuscript is now acceptable for publication, you may indicate that here to bypass the “Comments to the Author” section, enter your conflict of interest statement in the “Confidential to Editor” section, and submit your "Accept" recommendation.

Reviewer #1: All comments have been addressed

Reviewer #2: (No Response)

Reviewer #3: (No Response)

2. Is the manuscript technically sound, and do the data support the conclusions?

Reviewer #1: Yes

Reviewer #2: Yes

Reviewer #3: Partly

3. Has the statistical analysis been performed appropriately and rigorously? 

Reviewer #1: Yes

Reviewer #2: Yes

Reviewer #3: No

4. Have the authors made all data underlying the findings in their manuscript fully available?

Reviewer #1: Yes

Reviewer #2: Yes

Reviewer #3: Yes

5. Is the manuscript presented in an intelligible fashion and written in standard English?

Reviewer #1: Yes

Reviewer #2: Yes

Reviewer #3: Yes

6. Review Comments to the Author

Reviewer #1: All comments have been adressed properly and most of necessary changes have been made in relation with present possibilities for making them.

Reviewer #2: The authors have dealt with the various queries in a satisfactory manner and remaining comments are minor.

L80 currently reads “..differences in maternal recognition and early pregnancy in humans compared to cattle”. I suggest changing this to “differences in the maternal recognition of pregnancy and the timing and mode of implantation…

The description of the explant methodology is improved. One further point, if the explants weighed on average 42 mg but <20 mg was used to extract, presumably a piece was cut off prior to extraction. Please explain.

Lines 236-238. The accuracy of the variation calculation from the PCA plots does not justify giving the outcomes to 2 decimal places.

Reviewer #3: On lines 193 and 198 the authors indicated: “the B. taurus genome used as the statistical background”

A critical aspect of in silico analysis of gene function is the choice of genes to compose the background list. See papers listed below for a reference as to how using the wrong list of genes, such as all genes in a given genome, can produce biased results. This problem must be addressed.

https://genomebiology.biomedcentral.com/articles/10.1186/s13059-015-0761-7

https://bmcbioinformatics.biomedcentral.com/articles/10.1186/s12859-017-1571-6

https://www.nature.com/articles/s41596-018-0103-9.pdf?proof=true19

7. PLOS authors have the option to publish the peer review history of their article (what does this mean?). If published, this will include your full peer review and any attached files.

Reviewer #1: No

Reviewer #2: No

Reviewer #3: No

---

## [Author Response · Author response to Decision Letter 1]

25 Sep 2020

The authors would like to express our sincere thanks to the editor and the reviewers for taking the time to read our manuscript and for helping us to make it better – your work is very much appreciated. 

Reviewer #1

All comments have been addressed properly and most of necessary changes have been made in relation with present possibilities for making them: We thank the reviewer for your time assessing the revisions and are pleased to hear that we have addressed your comments properly. 

Reviewer #2

L80 currently reads “..differences in maternal recognition and early pregnancy in humans compared to cattle”. I suggest changing this to “differences in the maternal recognition of pregnancy and the timing and mode of implantation: Line 80-81: amended as indicated 

The description of the explant methodology is improved. One further point, if the explants weighed on average 42 mg but <20 mg was used to extract, presumably a piece was cut off prior to extraction. Please explain.: Line 131: Amended to make clear that <20 mg was cut off each explant that RNA was extracted from whilst still frozen, using sterile scissors.

Lines 236-238. The accuracy of the variation calculation from the PCA plots does not justify giving the outcomes to 2 decimal places. :Lines 236-238: Amended to 1 decimal place in the text.

Reviewer #3

On lines 193 and 198 the authors indicated: “the B. taurus genome used as the statistical background”

A critical aspect of in silico analysis of gene function is the choice of genes to compose the background list. See papers listed below for a reference as to how using the wrong list of genes, such as all genes in a given genome, can produce biased results. This problem must be addressed.:

The data have been reanalysed in String v11.0 using the genes used in the DESeq analysis as the statistical background. As a result of this there were some changes in P values and genes included in KEGG pathways. Furthermore, there were more KEGG pathways over-represented with DEG. As such, the following amendments have been made:

• Line 29: abstract updated

• Line 193 and 197, String version changed. 

• Line 193: change in the statistical background definition. 

• Line 281-292: amended Go analysis section to reflect re-analysis

• Line 294-338: amended KEGG analysis section to reflect re-analysis, including updating Table 2 and S2 Table and amending Fig2 to reflect genes included in new analysis.

• Line 409-430: added in reference to IL-17 and MAPK signalling pathways to further support the results.

• Lines 341-357 and Fig 4 amended to reflect updates to String (reanalysed in version 11 to match the KEGG and GO category analysis). However, it is not possible with STRING to generate a protein interaction network with the DEseq analysis gene list. I have queried this with STRING and received the following explanation from Damian Szklarczyk: “If you do not input the background yourself, the background is assumed to be the whole STRING proteome. Because it is the default, we have pre-computed all the necessary numbers, and different combinations of cut-off and channels (2504 combinations per species). Then we compare it to the distribution of links within your input and compare these two to generate the p-value. If you provide your own background we would have to recompute the background distribution of PPI, each time your query, which would take a lot of resources. The cut-off below we compute it is 8000 proteins.” 

Therefore, with our background of >15,000 proteins, we are unable to use this to assess the protein interaction network. “

---

## [Decision Letter · Decision Letter 2]

11 Nov 2020

Interaction of preimplantation factor with the global bovine endometrial transcriptome

PONE-D-20-09982R2

Dear Dr. Wonfor,

We’re pleased to inform you that your manuscript has been judged scientifically suitable for publication and will be formally accepted for publication once it meets all outstanding technical requirements.

Kind regards,

Juan J Loor

Academic Editor

PLOS ONE

Additional Editor Comments (optional):

Reviewers' comments:

Reviewer's Responses to Questions

**Comments to the Author**

1. If the authors have adequately addressed your comments raised in a previous round of review and you feel that this manuscript is now acceptable for publication, you may indicate that here to bypass the “Comments to the Author” section, enter your conflict of interest statement in the “Confidential to Editor” section, and submit your "Accept" recommendation.

Reviewer #1: (No Response)

Reviewer #2: All comments have been addressed

Reviewer #3: All comments have been addressed

2. Is the manuscript technically sound, and do the data support the conclusions?

Reviewer #1: (No Response)

Reviewer #2: (No Response)

Reviewer #3: Yes

3. Has the statistical analysis been performed appropriately and rigorously? 

Reviewer #1: (No Response)

Reviewer #2: (No Response)

Reviewer #3: Yes

4. Have the authors made all data underlying the findings in their manuscript fully available?

Reviewer #1: (No Response)

Reviewer #2: (No Response)

Reviewer #3: Yes

5. Is the manuscript presented in an intelligible fashion and written in standard English?

Reviewer #1: (No Response)

Reviewer #2: (No Response)

Reviewer #3: Yes

6. Review Comments to the Author

Reviewer #1: (No Response)

Reviewer #2: None. According to your information under point 1 above, I do not need to put anything here but it wont let me submit it without doing so

Reviewer #3: Thank you for the revisions. They addressed the concerns I presented. No further changes are requested .

7. PLOS authors have the option to publish the peer review history of their article (what does this mean?). If published, this will include your full peer review and any attached files.

Reviewer #1: No

Reviewer #2: No

Reviewer #3: No

---

## [Editor Report · Acceptance letter]

24 Nov 2020

PONE-D-20-09982R2 

Interaction of preimplantation factor with the global bovine endometrial transcriptome 

Dear Dr. Wonfor:

I'm pleased to inform you that your manuscript has been deemed suitable for publication in PLOS ONE. Congratulations! Your manuscript is now with our production department. 

Kind regards, 

on behalf of

Dr. Juan J Loor 

Academic Editor

PLOS ONE